# Bioleaching of Sulfide Minerals by *Leptospirillum ferriphilum* CC from Polymetallic Mine (Armenia)

Arevik Vardanyan [1,2,*], Anna Khachatryan [1], Laura Castro [3], Sabine Willscher [4], Stoyan Gaydardzhiev [2], Ruiyong Zhang [5,*] and Narine Vardanyan [1]

1 Department of Microbiology, SPC "Armbiotechnology" of the National Academy of Sciences of Armenia, 14 Gyurjyan Str., Yerevan 0056, Armenia

2 GeMMe—Minerals Engineering, Materials & Environment, University of Liege, Allée de la Découverte 9, Sart-Tilman, 4000 Liège, Belgium

3 Department of Chemical and Materials Engineering, Complutense University of Madrid, 28040 Madrid, Spain

4 Faculty of Natural Sciences I, Martin-Luther-Universität Halle-Wittenberg, Hoher Weg 8, 06120 Halle, Germany

5 Key Laboratory of Marine Environmental Corrosion and Biofouling, Institute of Oceanology, Chinese Academy of Sciences, No. 7 Nanhai Road, Qingdao 266071, China

* Correspondence: arevik.vardanyan@asnet.am (A.V.); ruiyong.zhang@qdio.ac.cn (R.Z.); Tel.: +374-94900931 (A.V.); +86-(0)53282898851 (R.Z.)

**Abstract:** A strain of *Leptospirillum* sp. CC previously isolated from Akhtala polymetallic ore (Armenia) was studied. The main morphological and physiological characteristics of CC were revealed. The optimal growth temperature was 40 °C and optimal pH 1.5. A phylogenetic analysis based on 16S rRNA gene sequences (GenBank ID OM272948) showed that isolate CC was clustered with *L. ferriphilum* and possessed 99.8% sequence similarity with the strain *L. ferriphilum* OL12-2 (KF356024). The molar fraction of DNA (G + C) of the isolate was 58.5%. Bioleaching experiment indicates that *L. ferriphilum* CC can oxidize Fe(II) efficiently, and after 17 days, 44.1% of copper and 91.4% of iron are extracted from chalcopyrite and pyrite, respectively. The efficiency of *L. ferriphilum* CC in pyrite oxidation increases 1.7 times when co-cultivated with *At. ferrooxidans* ZnC. However, the highest activity in pyrite oxidation shows the association of *L.ferriphilum* CC with heterotrophic *Acidocella* sp. RBA bacteria. It was shown that bioleaching of copper and iron from chalcopyrite by association of *L. ferriphilum* CC, *At. ferrooxidans* ZnC, and *At. albertensis* SO-2 in comparison with pure culture *L. ferriphilum* CC for 21 days increased about 1.2 and 1.4–1.6 times, respectively.

**Keywords:** *Leptospirillum ferriphilum*; isolation; characterization; phylogenetic analysis; bioleaching

## 1. Introduction

Although *At. ferrooxidans* was considered to be the most important microorganism in bioleaching of metals for many years, leptospirilla have been found to be the dominant iron-oxidizing bacteria in gold–arsenopyrite and pyrite biooxidation reactors operating at 40 °C [1–6]. One of the major factors that determines the dominance of particular microorganisms in the commercial bioleaching operations is the ratio of ferric/ferrous ions (related to the redox potential). In contrast to *At. ferrooxidans*, *Leptospirillum ferrooxidans* is found to be resistant even to 500 mM ferric iron concentration [7]. Both *L. ferrooxidans* and *L. ferriphilum* can oxidize ferrous iron even at a pH as low below 1.0 and up to 40 °C temperature [6,8–11].

On the basis of 16S rRNA gene phylogeny, the genus *Leptospirillum* has been divided into three groups [12]. At present, the genus *Leptospirillum* comprises four species of Gram-negative obligately aerobic chemolithotrophic bacteria: *Leptospirillum ferrooxidans* (group I) [13], *Leptospirillum rubarum* (group II) [14], *Leptospirillum ferriphilum* and *L. ferrodiazotrophum* (group III) [14,15]. In addition, microbial community genomics has identified further species of "*Leptospirillum* sp. group IV UBA BS" [14,16].



*Leptospirillum* spp. are vibrio and spiral-shaped chemolithotrophic organisms that fix carbon ($CO_2$) using Fe(II) as energy source [1,17–19].

The temperature optimum is between 30 to 37 °C, although many isolated strains are defined as being moderately thermophilic (above 40 °C). *Leptospirillum* spp. bacteria are able to grow in a pH range from 1.0–2.0 [20].

*Leptospirillum ferriphilum* belongs to the bioleaching microbial communities involved in solubilization of metals from sulfide ores [21].

As reported by Galleguillos et al. (2009) [22], the metal resistance ability of the *L. ferriphilum* was far greater than *At. ferrooxidans* and *L. ferrooxidans*, which makes it an ideal candidate for bioreoxidation of ferrous iron from leachates containing different metals. The organism is quite useful in two-stage bioleaching processes for treatment of copper and other base metal concentrates [23–26].

Recently, scientists particularly highlighted species of *Leptospirillum* genera and their mixed cultures with other bacteria in the processes of biooxidation and bioleaching of minerals occurring at temperatures higher than 40 °C [17,27–31]. *Leptospirillum* spp. bacteria capable of oxidizing Fe(II) within the temperature range of 30–40 °C have been isolated from leaching pulps of copper concentrate and arsenopyrite concentrates.

This study addresses the characterization and reclassification based on 16S rRNA analysis of the previously isolated and described strain *L. ferriphilum* CC dominate in a bioleaching pulp of copper concentrate [32]. The objective of this study is to assess the potential of the isolated strain as a promising candidate for the regeneration of ferric iron and biodegradation of sulfide minerals.

## 2. Materials and Methods

### 2.1. Cultures and Growing Conditions

In this study, *L. ferriphilum* CC, iron- and sulfur-oxidizing *At. ferrooxidans* ZnC, sulfur-oxidizing *At. albertensis* SO-2 (KP455986), and heterotrophic *Acidocella* sp. RBA (KX784767) isolated by us previously were used [32–35]. Before bioleaching experiments, *L. ferriphilum* CC and *At. ferrooxidans* ZnC were grown on MAC medium [36] with ferrous iron as a source of energy at 40 and 30 °C, respectively, *At. albertensis* SO-2 was grown on MAC medium with elemental sulfur as a source of energy at 30 °C, and *Acidocella* sp. RBA was grown on LHET2 35 °C [37].

### 2.2. Morphology and SEM Images

Gram-staining was performed by Huker method [38] and was observed with Leica DM500 trinocular (×1000) microscope. For SEM studies, bacterial culture *L. ferriphilum* CC was grown on MAC medium at pH 3.0 containing ferrous iron at 40 °C. Then bacterial culture was filtered onto a 0.2 μm pore-size membrane, then the sample was successively dehydrated with acetone/water mixtures of 30%, 50%, and 70% acetone, and stored overnight at 4 °C in 90% acetone. The sample was dried by critical-point drying and coated with gold. A Zeiss Sigma 300V P FEG scanning electron microscope operating at 5 kV was used to observe samples.

### 2.3. Optimal pH and Temperature for Growth

The study of the effect of temperature and pH on the growth of CC strain was carried out in 100 mL flasks containing 50 mL of sterile MAC medium, 5 mL inoculum on the rotary shaker, and cultures were agitated at 150 rpm. Growth ranges for temperature and pH were set as 25–50 °C and 0.5 to 2.5, respectively.

### 2.4. Determination of Organic Acids in Culture Liquid of L. ferriphilum CC

Organic acids in the culture liquid were determined by HPLC. The method is based on the use of reverse-phase high-performance liquid chromatography (HPLC). The mass concentration (mass fraction) of organic acids in the sample was determined by a diode array detector. Eluent: phosphate buffer solution, molar concentration 0.1 mol/dm³, pH

2.2–2.6, column temperature−30 °C, ambient temperature −22 ± 1 °C. The measurements were carried out in the wavelength range of the diode array detector: 200–600 nm. Eluent flow rate: 1 ml/min, injected sample volume −5–10 μl.

Prior to analysis, the samples were preconcentrated under a stream of nitrogen in a NER-13 nitrogen concentrator, the concentrated samples were passed through a membrane filter with a pore diameter of 0.95 μm, and then through a C18 purification column.

### 2.5. Phylogenetic Analysis

Total DNA of CC strain was extracted by following a protocol provided by the NucleoSpin Microbial DNA Kit (Macherey-Nagel, Düren, Germany). Polymerase chain reaction (PCR) was performed to amplify the 16S rRNA gene region by using the genomic DNA of the strains as template, universal bacterial primers fD1 (27F) (AGAGTTTGATCCTG-GCTCAG) and rP2 (ACGGCTACCTTGTTACGAG) as primers. PCR products were tested by 1.5% agarose gel electrophoresis and sequenced with primers 908fwd (16Sfwd) (GT-GCCAGCAGCCGCG) and 796rev (16Srev) (GGGTTGCGCTCGTTG) by Microsynth AG (Balgach, Switzerland). Close relative and phylogenetic affiliation of the obtained 16S rRNA sequences were determined by submitting to the NCBI 16S ribosomal RNA GenBank database using NCBI BLAST search analyses (www.ncbi.nlm.nih.gov) performed with Geneious prime 2022.0.2. (https://www.geneious.com) and the 16S Biodiversity tool (RDP tool version 2.12) [39,40]. Construction of phylogenetic trees was performed by MEGA X software using neighbor-joining method [41,42].

DNA base composition (G + C) content was determined using HPLC Method [43].

### 2.6. Leaching Experiments

Pyrite ($FeS_2$) and chalcopyrite ($CuFeS_2$) from Shamlugh ore deposit (Armenia) were tested in the bioleaching experiments. Chemical composition of minerals is presented in Table 1. Feed minerals were ground to a particle size ≤ 63 μm.

**Table 1.** Chemical composition of the analyzed minerals (wt%).

| Sample | Fe | Cu | S |
|:---:|:---:|:---:|:---:|
| Pyrite | 43.8 | - | 49.0 |
| Chalcopyrite * | 29.7 | 30.2 | 33.8 |

* It can be supposed that tested sample of chalcopyrite may contain some amount of enargite ($Cu_3AsS_4$).

Bioleaching of pyrite and chalcopyrite was performed using pure culture of *L. ferriphilum* CC as well as its associations with *At.ferrooxidans* ZnC, *At. albertensis* SO-2, and heterotrophic bacteria *Acidocella* sp. RBA [34]. Bioleaching experiments with *L. ferriphilum* CC were carried out in 250 mL Erlenmeyer flasks containing 100 mL of MAC medium without iron at 40 °C. Comparative studies on bioleaching of minerals by pure and mixed cultures were performed at 35 °C, which is closer to optimal growth temperatures of all bacteria used. Pulp density (PD) was 4% and pH 1.5. The inoculum of used cultures was 5% and all experiments were carried out in triplicate. For each bioleaching experiment chemical controls with the same conditions and without inoculum were included. Copper, total iron, ferric (Fe(III)), and ferrous (Fe(II)) ions in leachate were analyzed at 24 h intervals and pH was recorded as well. The redox potential was measured with an oxidation/reduction potentials (ORP) electrode met BNC-connector (Pt/Ag/AgCl) of Hi2211-01 Benchtop pH/mV Meter (Hanna Instruments, Vöhringen, Germany). pH was determined with a Hi2211-01 Benchtop pH/mV Meter equipped with an Ag/AgCl electrode. Copper and total iron were determined by atomic-absorption spectrophotometer AAS 1N (Carl Zeiss, Jena, Germany) using an air–propane–butane flame. Concentrations of ferric (Fe(III)) and ferrous (Fe(II)) ions were determined by the complexometric method with EDTA [44].

## 3. Results

### 3.1. Cell Morphology

Cells of *Leptospirillum* sp. CC are Gram-negative and are motile, vibrio- or spiral-shaped (Figure 1, Supplementary Figure S1). They have a diameter of 0.2–0.6 μm and a length of 1.2–1.9 μm.

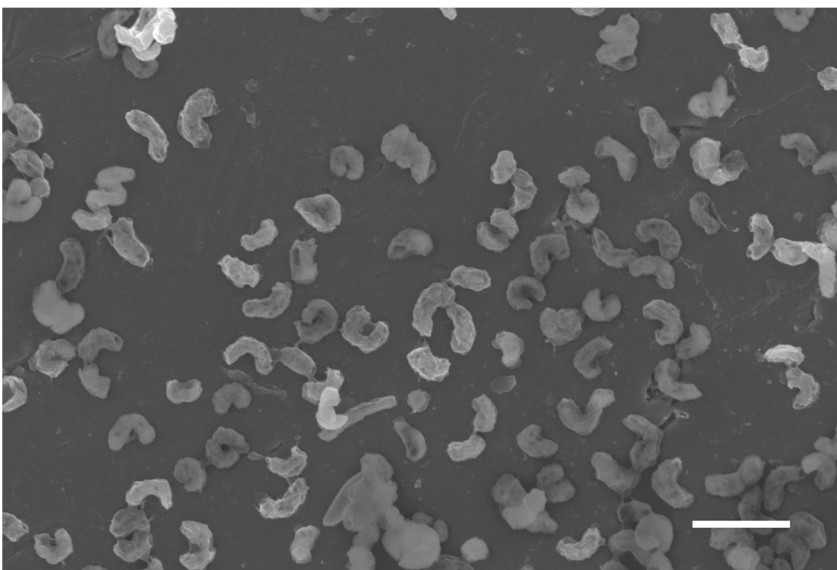

**Figure 1.** SEM micrograph of *Leptospirillum* sp. CC. Cells were grown on ferrous iron at 40 °C for 5 days. Bar represents 2 μm.

This observation is typical for the genus of *Leptospirillum*. Cells were mainly vibrio-like, and seldom were spiral with up to ten turns (Supplementary Figure S1). The morphology results show that strain CC is consistent with previously described *Leptospirillum* species [45].

### 3.2. Physiological–Biochemical Characteristics

Optimal pH and temperature for growth: the study on the influence of temperature and pH on the growth of CC strain indicates an optimum of 40 °C and pH 1.5, respectively (Figure 2). No growth is detected at 50 °C. pH 0.5 is the lower limit for the growth of CC strain (Figure 2).

The influence of $Fe^{2+}/Fe^{3+}$, as well as Cu, Zn, Ni, and Co ions, on the growth of *Leptospirillum* sp. CC and $Fe^{2+}$ oxidation was studied previously. The comparison of iron oxidation kinetic parameters of *Leptospirillum* sp. CC with other strains of *L. ferriphilum* indicates the high potential of *Leptospirillum* sp. CC strain in view of biogenic regeneration of concentrated ferric iron ($Fe^{3+}$) during the bioleaching processes of ores and mineral concentrates [32].

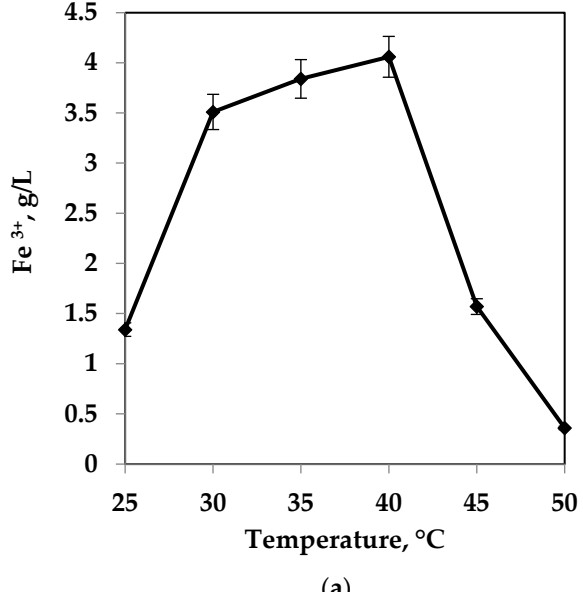

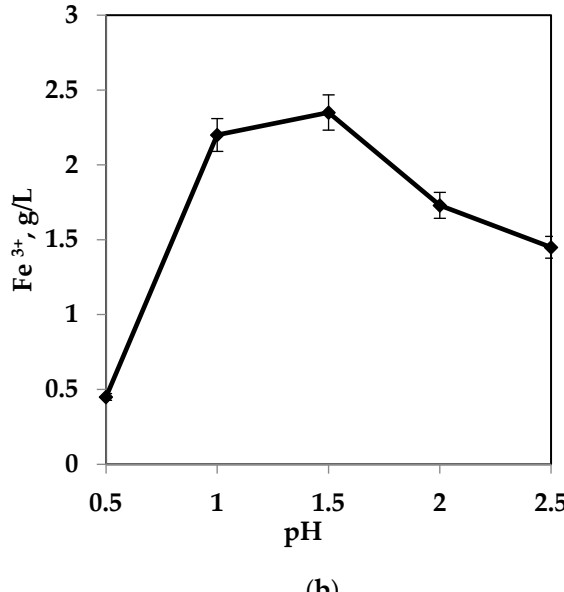

(**a**)　　　　　　　　　　　　　　　　　　　　　(**b**)

**Figure 2.** Effect of temperature (**a**) at pH 1.6 and pH (**b**) at temperature 40 °C on iron oxidation by *L. ferriphilum* CC.

### 3.3. Phylogenetic Analysis of 16S RNA

The sequence of 16S rRNA of *Leptospirillum* sp. CC was submitted to the GenBank and the accession number OM272948 was obtained. Based on homology of 16S rRNA, the phylogenetic tree was built as shown in Figure 3. Isolate CC was clustered with *Leptospirillum ferriphilum* strains and possessed 99.80% sequence similarity with *L. ferriphilum* OL 12-2 (Figure 3, Table 2). In Figure 3, *Lactobacillus acidophilus* is used as an outgroup to root the tree, and the database accession numbers of the gene sequences used are given in parentheses.

**Table 2.** Identity of 16S rRNA of isolated *Leptospirillum* sp. CC with other *L. ferriphilum* strains.

| Isolated Strain | Type Strains (Accession Numbers) | Identity, % | Reference |
|---|---|---|---|
| **L. ferriphilum CC** | *L. ferriphilum* OL 12-2 (KF356024.1) | 99.80 | Moshchanetskiy et al., 2014 [48] |
| | *L. ferriphilum* MP1 (MN780596.1) | 99.73 | Muravyov and Panyushkina, 2020 [49] |
| | *L. ferriphilum* P1 (MG386692.1) | 99.60 | Panyushkina et al., 2018 [50] |
| | *L. ferriphilum* P3a (NR028818.1) | 98.27 | Coram and Rawlings, 2002 [51] |

Chromosomal DNA base analysis shows that *Leptospirillum* sp. CC has a G + C content of 58.5%, which is close to, or coincides with, Group II Leptospirilla capable of growing in the temperature range of 35−45 °C [30]. Thus, G + C content analysis suggests that strain CC belongs to Group II Leptospirilla.

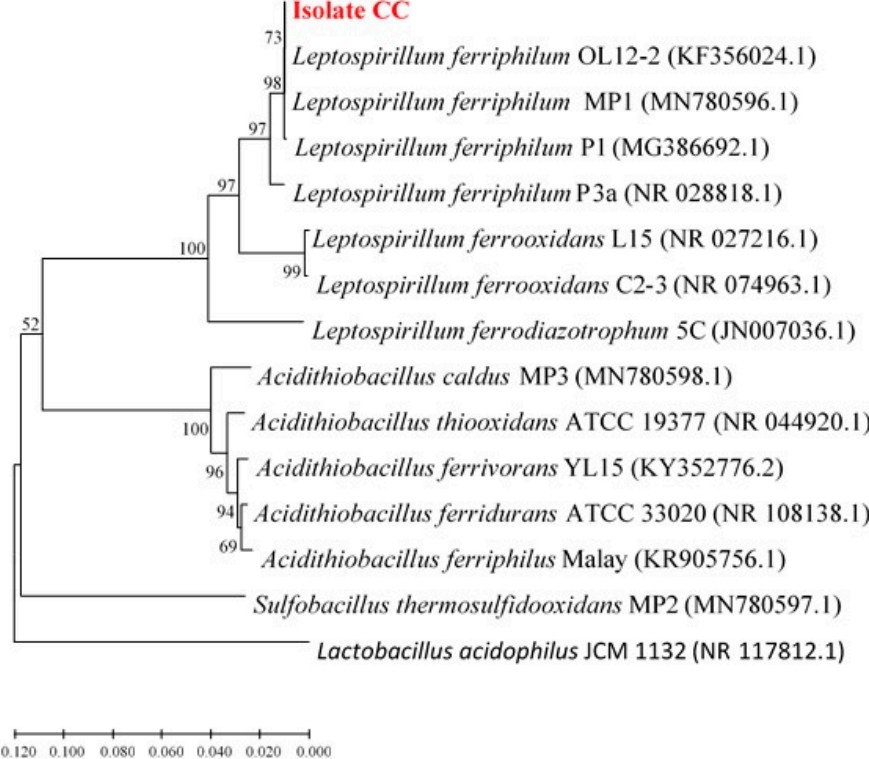

**Figure 3.** Phylogenetic position of strain CC. The evolutionary history was inferred using the neighbor-joining method [46]. The percentage of replicate trees in which the associated taxa clustered together in the bootstrap test (10,000 replicates) are shown next to the branches [47].

### 3.4. Leaching Results

Bioleaching of pyrite and chalcopyrite: the bioleaching trends for pyrite and chalcopyrite by *L. ferriphilum* CC at 40 °C are shown in Figures 4 and 5, respectively. As shown in Figure 4, *L. ferriphilum* CC demonstrates an elevated pyrite-oxidizing activity. After 17 days of bioleaching, 26.3% of iron was leached resulting in 4.6 g/L concentration in the leached solution.

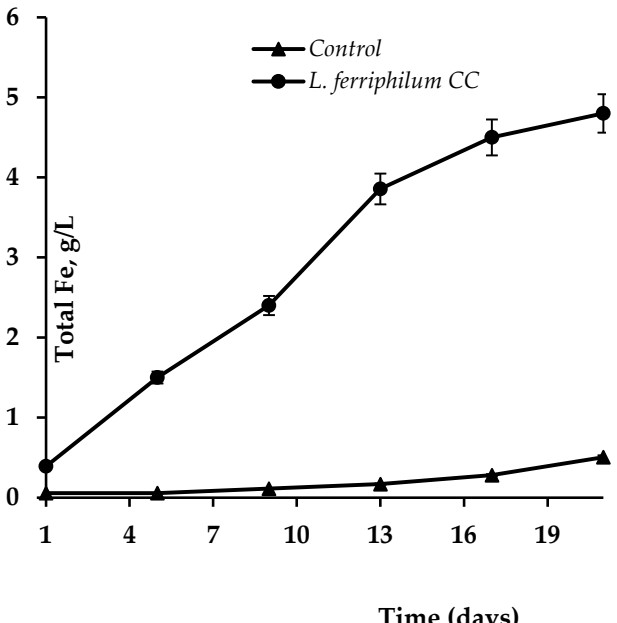

**Figure 4.** Bioleaching of pyrite by *L. ferriphilum* CC (T 40 °C; pH 1,5; PD 4%).

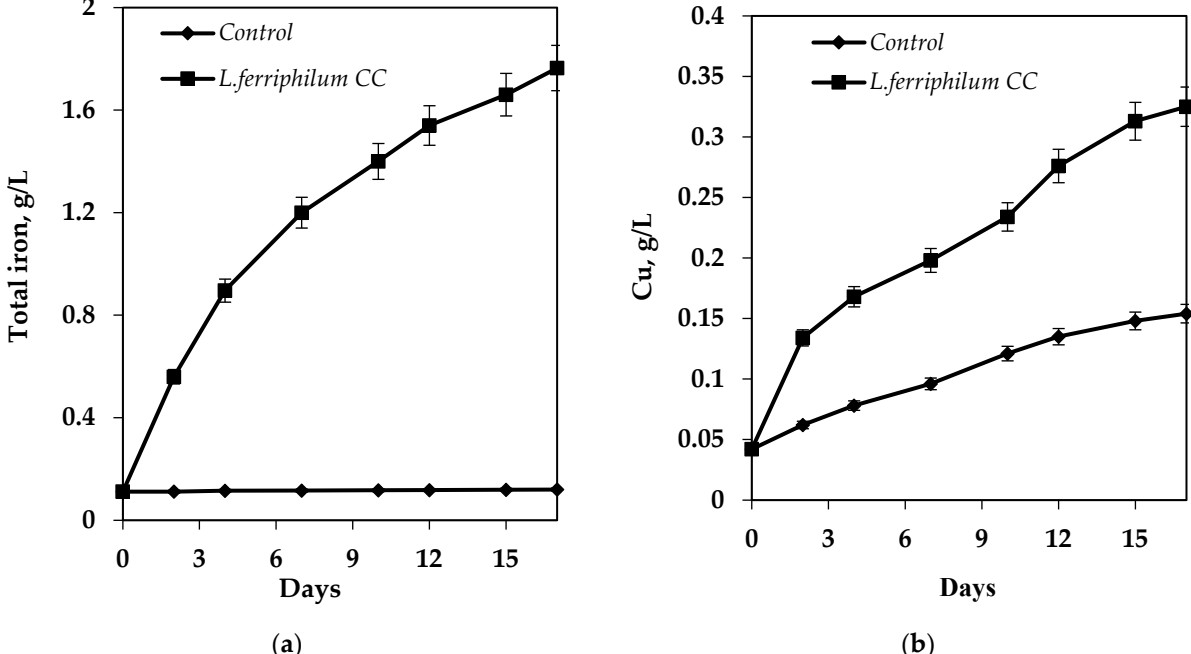

**Figure 5.** Bioleaching of iron (**a**) and copper (**b**) from chalcopyrite by *L. ferriphilum* CC (T 40 °C; pH 1.5, PD 4%).

As shown in Figure 5, the amount of leached copper increases with the bioleaching time and reaches 0.325 g/L corresponding to 3% for 15 days. Thus, activity of *L. ferriphilum* CC in biodegradation of chalcopyrite is much lower compared with pyrite.

### 3.4.1. Pyrite Leaching by *L. ferriphilum* CC with Associations of Other Iron- and Sulfur-Oxidizing and Heterotrophic Bacteria

The operational feasibility of the bioleaching and biooxidation processes is largely determined by the nature of the used microorganisms. It has been shown that associations and natural consortia of microorganisms function in a more efficient and stable way in commercial bioleaching installations than the corresponding pure cultures [52–56]. Therefore, the development and establishment of a highly active resistant microbial associations for use at commercial scale remains an important challenge.

Based on the above-mentioned, the associations of *L. ferriphilum* CC with iron- and sulfur-oxidizing bacteria *At. ferrooxidans* ZnC, sulfur-oxidizing bacteria *At. albertensis* SO-2, and heterotrophic bacteria *Acidocella* sp. RBA were studied.

The data presented in Figure 6 show that the efficiency of *L. ferriphilum* CC in pyrite oxidation increases by 1.7 times when co-cultivated with *At. ferrooxidans* ZnC. *At. albertensis* SO-2 in association with *L. ferriphilum* has no significant effect (1.2 times) on pyrite bioleaching. However, the association constructed on the basis of *L. ferriphilum* CC and *Acidocella* sp. RBA bacteria makes it possible to increase the amount of total iron leached from pyrite by about 2.8 times (Figure 6). Thus, the association of *L. ferriphilum* CC with heterotrophic bacteria shows the highest activity in pyrite oxidation (Figure 6).

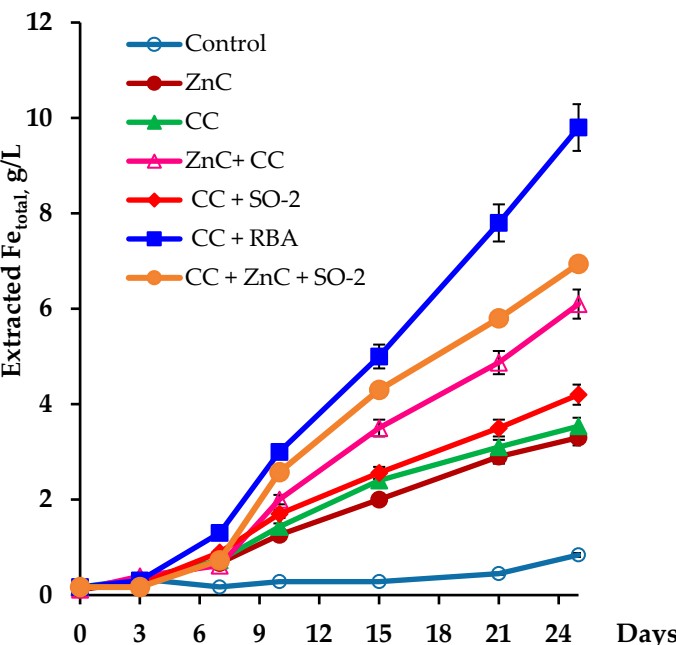

**Figure 6.** Bioleaching of pyrite by mono and mixed cultures of *L. ferriphilum* CC, *At. ferrooxidans* ZnC, *At. albertensis* SO-2, and *Acidocella* sp. RBA (T 35 °C; pH 1.8; 180rpm; PD 4%).

It should be noted that when using a pure culture of *At. ferrooxidans* ZnC, leached iron is present in the ferric and ferrous form in approximately equal amounts. Accordingly, the redox potential of the leaching solution differs slightly from that of the non-inoculated control (Table S1). A low ORP (625 mV) is also observed when using the association of *L. ferriphilum* CC with the sulfur-oxidizing bacterium *At. albertensis* SO-2. When using associations of *L. ferriphilum* CC with *At. ferrooxidans* ZnC, due to the high iron-oxidizing activity of *L. ferriphilum* CC, the leached iron is exclusively in the form of ferric iron, which provides the highest ORP (850 mV) value and, therefore, a high oxidizing power of the leaching solution. Despite the fact that when using *L. ferriphilum* CC with *Acidocella* sp. RBA, the highest amount of leached iron is observed, the ORP of the solution is significantly lower, at −715 mV (Table S1). Thus, the ORP, and, consequently, the oxidizing features of the leaching solution, depend on the activity of the iron-oxidizing bacteria or the associations between them.

The degree of iron extraction by the association of *L. ferriphilum* CC and *Acidocella* sp. RBA reach 56%, while in the case of the monoculture of *L. ferriphilum* CC and *At. ferrooxidans* ZnC, this parameter does not exceed 20%.

Pyrite is insoluble in acid and, therefore, according to the mechanism of oxidation of sulfide minerals, can be dissolved only under the action of ferric iron. Thus, the presence of *L. ferriphilum* CC in association with *At. ferrooxidans* ZnC leads to intensive oxidation of Fe(II) ions and regeneration of ferric iron (Fe(III)), which, in turn, accelerates the oxidation of pyrite according to the equation below (Equation (1)):

$$FeS_2 + 7Fe_2(SO_4)_3 + 8\,H_2O \rightarrow 15\,FeSO_4 + 8H_2SO_4 \tag{1}$$

### 3.4.2. Bioleaching of Chalcopyrite by *L. ferriphilum* CC with Associations of *At. ferrooxidans* ZnC, *At. albertensis* SO-2, and *Acidocella* sp. RBA

The data shown in Figures 7 and 8 indicate that associations of iron-oxidizing bacterium *L. ferriphilum* CC with *At. ferrooxidans* ZnC and sulfur-oxidizing *At.albertensis* SO-2 oxidize chalcopyrite much more actively than a pure culture of *L. ferriphilum* CC. Thus, in the presence of *At. ferrooxidans* ZnC and *At. albertensis* SO-2, the leaching of copper and iron by *L. ferriphilum* CC from chalcopyrite increases, approximately 1.2 and 1.4–1.6 times, respectively (Figures 7 and 8). It is noteworthy that in the leaching of copper and iron from

chalcopyrite, the association of *L. ferriphilum* CC with a heterotrophic bacterium seems to be more effective in comparison with pure culture (1.3 and 1.85 times, respectively). However, the association consisting of *L. ferriphilum* CC, *At. ferrooxidans* ZnC, and sulfur-oxidizing bacterium *At. albertensis* SO-2 shows the highest efficiency in leaching of copper and iron from chalcopyrite (Figures 7 and 8). This is because for 25 days of bioleaching of chalcopyrite with the association of *L. ferriphilum* CC, *At. ferrooxidans* ZnC, and *At. albertensis* SO-2, copper extraction reaches 15% and iron extraction 33% (Table 3).

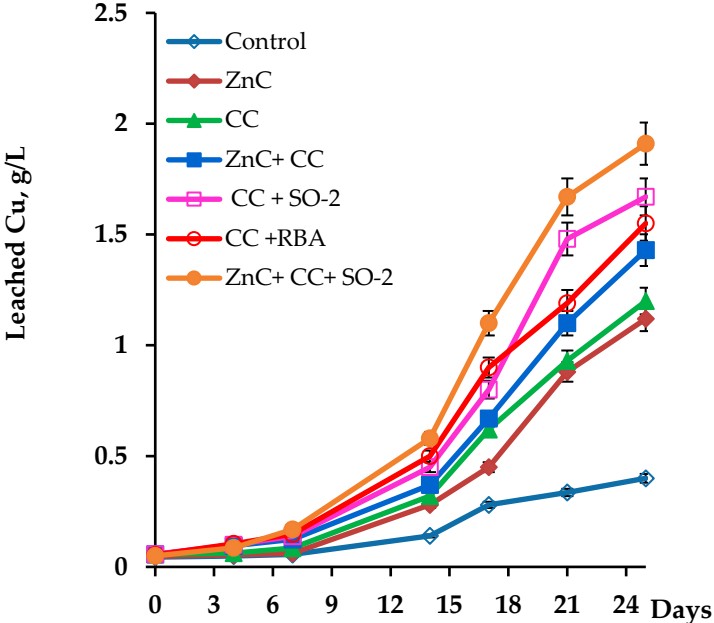

**Figure 7.** Leaching of copper from chalcopyrite by a pure culture of *L. ferriphilum* CC and association with other iron- and sulfur-oxidizing bacteria (T 35 °C; pH 1.8; 180 rpm; PD 4%).

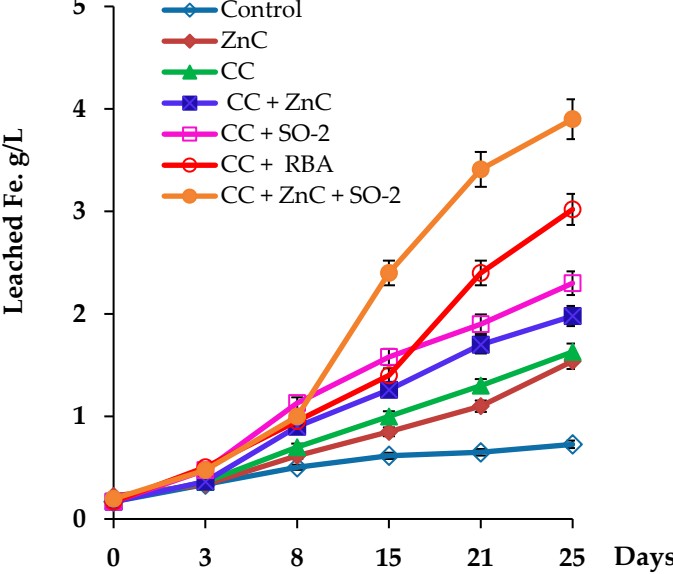

**Figure 8.** Leaching of iron from chalcopyrite by a pure culture of *L. ferriphilum* CC and association with other iron- and sulfur-oxidizing bacteria and *Acidocella* sp. RBA heterotrophic bacteria (T 35 °C; pH 1.8; 180 rpm; PD 4%).

**Table 3.** Leaching of iron and copper from chalcopyrite by *L. ferriphilum* CC and association with iron- and sulfur-oxidizing and heterotrophic bacteria.

| Bacteria and Their Associations | Extracted for 25 Days | | | | | | Final | |
| --- | --- | --- | --- | --- | --- | --- | --- | --- |
| | Iron | | | | Copper | | | |
| | g/L | | Fe total | % Fe total | g/L | % | pH | ORP, mV |
| | $Fe^{3+}$ | $Fe^{2+}$ | | | | | | |
| Control | 0 | 0.672 | 0.672 | 5.6 | 0.44 | 3.5 | 1.8 | 520 |
| *L. ferriphilum* CC | 1.4 | 0.23 | 1.63 | 13.6 | 1.2 | 9.4 | 1.8 | 640 |
| *At. ferrooxidans* ZnC | 1.096 | 0.448 | 1.544 | 12.8 | 1.12 | 8.7 | 1.75 | 600 |
| *L. ferriphilum* CC + *At. ferrooxidans* ZnC | 2.632 | 0.112 | 2.632 | 22.1 | 1.43 | 11.2 | 1.7 | 680 |
| *L. ferriphilum* CC + *At. albertensis* SO-2 | 2.016 | 0.616 | 2.744 | 23.0 | 1.67 | 13.4 | 1.6 | 720 |
| *L. ferriphilum* CC + *Acidocella* sp. RBA | 0.336 | 3.024 | 3.360 | 28.2 | 1.55 | 12.1 | 1.6 | 620 |
| *L. ferriphilum* CC + *At. ferrooxidans* ZnC + *At. albertensis* SO-2 | 3.960 | 0 | 3.960 | 33.3 | 1.91 | 14.9 | 1.5 | 815 |

The data in Table 3 show that chalcopyrite bioleaching is correlated with changes in solution pH and ORP. When using *L. ferriphilum* CC and *At. ferrooxidans* ZnC as monoculture, the final pH is 1.8 and 1.75, and the ORP is 640 and 600 mV, respectively, while in case of *L. ferriphilum* CC in association with *At. albertensis* SO-2, the pH is relatively lower (1.6), and the ORP is significantly higher (720 mV). The lowest pH (1.5) and the highest ORP value (815 mV) are observed in case of using the association consisting of *L. ferriphilum* CC, *At. ferrooxidans* ZnC, and *At. albertensis* SO-2 (Table 3).

Chalcopyrite is an acid-soluble sulfide mineral and is, therefore, attacked by both ferric iron ($Fe^{3+}$) and protons ($H^+$) (Equations (2) and (3)) [57,58].

$$CuFeS_2 + 4H^+ \rightarrow Fe^{2+} + Cu^{2+} + 2H_2S \tag{2}$$

$$CuFeS_2 + 2Fe_2(SO_4)_3 \rightarrow CuSO_4 + 5FeSO_4 + 2S^0 \tag{3}$$

Ferric ions oxidize chalcopyrite, releasing copper and iron, as well as elemental sulfur, into solution (Equation (2)). The role of *L. ferriphilum* CC lies in the regeneration of the oxidizing agent—Fe(III) (Equation (4)).

$$Fe^{2+} + H^+ + 0.5O_2 \xrightarrow{At.ferrooxidans} Fe^{3+} + H_2O \tag{4}$$

It is assumed that iron-oxidizing bacteria accelerate the leaching of chalcopyrite through the ferric iron they produce. *At. albertensis* in a mixed culture oxidizes sulfide sulfur to sulfuric acid, and thereby prevents the formation of jarosite and the hydrophobic layer of sulfur on the surface of chalcopyrite (Equation (5)). Thus, it ultimately limits the mineral passivation phenomena and promotes the oxidation rate of chalcopyrite.

$$0.125S_8 + 1.5O_2 + H_2O \xrightarrow{At.ferrooxidans,At.caldus} SO_4^2 + 2H^+ \tag{5}$$

*3.5. Determination of Organic Acids in Culture Liquid of L. ferriphilum CC*

The data shown in Tables 4 and 5 have to be regarded in conjunction with the fact that both *At. ferrooxidans* and *Leptospirillum* excrete low molecular compounds (organic acids) to the solution. Excretion of some organic acids by the mentioned bacteria has been observed in our previous work [34].

**Table 4.** Analysis of organic acids in culture liquid *L. ferriphilum* CC by HPLC.

| N | Organic Acids | Exit Time | Peak Area | Height | Conc., g/L | Final Concentration, mg/L |
|---|---|---|---|---|---|---|
| 1 | Tartaric acid | 4.523 | 266,4847 | 80,296 | 1948.151 | 160.078 |
| 2 | Malic acid | 5.630 | 374,937 | 20,535 | 316.628 | 26.017 |
| 3 | Lactic acid | 7.200 | −506 | −7 | 0.000 | 0.000 |
| 4 | Acetic acid | 7.940 | −825 | −1 | 0.000 | 0.000 |
| 5 | Citric acid | 10.358 | 6377 | 394 | 2.316 | 0.19 |

**Table 5.** Analysis of organic acids in culture liquid *At. ferrooxidans* ZnC by HPLC.

| N | Organic Acids | Exit Time | Peak Area | Height | Concentration, g/L | Final Concentration, mg/L |
|---|---|---|---|---|---|---|
| 1 | Tartaric acid | 4.538 | 6,309,607 | 232,851 | 4612.673 | 230.6 |
| 2 | Malic acid | 5.657 | 562,506 | 42,482 | 475.027 | 23.75 |
| 3 | Lactic acid | 7.070 | −26,250 | −81 | 0.000 | 0.000 |
| 4 | Acetic acid | 8.481 | 5645 | 312 | 7.677 | 0.384 |
| 5 | Citric acid | 10.363 | 51,123 | 1937 | 18.569 | 0.928 |

Analysis of the culture liquid of *At. ferrooxidans* ZnC and *L. ferriphilum* CC after cell removal by high-performance liquid chromatography (HPLC) shows the presence of tartaric, malic, acetic, and citric acids. The results of the analysis of organic acids in samples of the culture liquid *At. ferrooxidans* ZnC and *L. ferriphilum* CC are presented in Tables 4 and 5.

Berthelot et al. [59]., Johnson and Roberto [60]., and Liu et al. [61] proposed that the presence of heterotrophs, such as *A. acidophilum*, could improve metal bioleaching by biodegrading organic matter and, thus, detoxify the growth environment for other acidophiles. Paiment et al. [62] in a 21 day bioleaching experiment, using *At. ferrooxidans* alone or mixed culture with *A. acidophilum*, showed that the recovery of copper from copper–nickel sulfide ore increased from 1.7 to 2.5% in case of mixed culture. Bacelar-Nicolau and Johnson [63] found that pyrite leaching was enhanced when mixed cultures containing iron oxidizers and *A. acidophilum* were used.

Thus, acidophilic heterotrophic *Acidocella* sp. RBA bacteria can utilize organic compounds contained in exudate or lysate of cells and, thus, reduce their toxic effect on autotrophic bacteria such as *At. ferrooxidans* ZnC and *L. ferriphilum* CC. In addition, heterotrophic *Acidocella* sp. RBA bacteria excrete $CO_2$ during respiration that can be assimilated by autotrophic bacteria in their constructive metabolism.

Thus, interactions between iron-oxidizing and sulfur-oxidizing autotrophs and heterotrophs in association increase the extraction of metals.

It is well-known that a mixed culture consisting of moderately thermophilic bacteria *L. ferrooxidans* and *At. caldus* leaches chalcopyrite more efficiently than the mesophilic bacterium *At. ferrooxidans* in pure and mixed culture. In addition, it was noted that when using *At. ferrooxidans*, passivation of the chalcopyrite surface and inhibition of mineral leaching quickly occurred [52,54,64–66].

## 4. Conclusions

The cells of CC are Gram-negative and are motile, vibrio- or spiral-shaped, with a 0.12–0.13 μm width and a 0.6–1.0 μm length. It has a guanine plus cytosine (G + C) content of 58.5% and exhibits 99.8% similarity of 16S rRNA to *L. ferriphilum* OL12-2.

Physiological investigation has indicated that *L. ferriphilum* CC is an obligate chemolithoautotroph, metabolizing ferrous iron and pyrite. Optimal growth temperature for *L. ferriphilum* CC was found to be 40 °C and the optimal pH -1.5.

It has been shown that during bioleaching experiments *L. ferriphilum* CC can oxidize Fe(II) efficiently, and after 17 days, 44.1% of copper and 91.4% of iron are extracted from chalcopyrite and pyrite, respectively.

When co-cultivated with *At. ferrooxidans* ZnC, the efficiency of *L. ferriphilum* CC in pyrite oxidation increases by 1.7 times. The association of *L. ferriphilum* CC with heterotrophic *Acidocella* sp. RBA resulted in highest activity in pyrite oxidation.

It has also been shown that bioleaching of copper and iron from chalcopyrite by association of *L. ferriphilum* CC, *At. ferrooxidans* ZnC, and *At. albertensis* SO-2 in comparison with pure culture *L. ferriphilum* CC for 21 days increases about 1.2 and 1.4–1.6 times, respectively.

Thus, it is supposed that heterotrophic *Acidocella* sp. RBA bacteria can utilize organic compounds contained in exudate or lysate of cells and, therefore, reduce their toxic effect on autotrophic bacteria such as *At. ferrooxidans* ZnC and *L. ferriphilum* CC.

This indigenous strain may contribute to the iron cycling as well as to the acid mine drainage patterns in the local area.

**Supplementary Materials:** The following supporting information can be downloaded at: https://www.mdpi.com/article/10.3390/min13020243/s1, Figure S1: SEM micrograph of strain *Leptospirillum* sp. CC after growth of 5 days; Table S1: Iron leaching from pyrite by a pure culture of *L. ferriphilum* CC and associations with other iron- and sulfur-oxidizing and heterotrophic bacteria.

**Author Contributions:** Conceptualization, A.V., N.V. and R.Z.; methodology, A.V., N.V., R.Z. and L.C.; software, A.V.; validation, S.W., L.C., S.G., N.V. and R.Z.; formal analysis, A.V., A.K., L.C., and R.Z.; investigation, A.V., A.K. and R.Z.; resources, N.V.; data curation, N.V. and R.Z.; writing—original draft preparation, A.V.; writing—review and editing, N.V., R.Z., L.C., S.W. and S.G.; visualization, A.V., N.V. and A.K.; supervision, N.V. and R.Z.; project administration, N.V.; funding acquisition, N.V. and A.V. All authors have read and agreed to the published version of the manuscript.

**Funding:** This work was supported by the Science Committee of RA, Research project № 22rl-031.

**Data Availability Statement:** Not applicable.

**Conflicts of Interest:** The authors declare no conflict of interest. The funders had no role in the design of the study; in the collection, analyses, or interpretation of data; in the writing of the manuscript; or in the decision to publish the results.

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
