# Peer review of "Bioleaching of Sulfide Minerals by Leptospirillum ferriphilum CC from Polymetallic Mine (Armenia)"

_minerals, doi:10.3390/min13020243_

Round 1

Reviewer 1 Report

The authors are thanked for their interesting contribution describing a new field isolate of Leptospirillum ferriphilum. There are some frequent issues with incorrect capitalization of the species designation ferriphilum through out the manuscript - this should be all lower case, not "Ferriphilum" (note that this error even appears in the title of the manuscript. Please review the text carefully to correct this. There is also inconsistent italicization of the binomial genus and species - for example, in the title and in multiple locations within the abstract (lines 25, 26) where the binomial should be presented in italics. See also line 194, lines 232-233 ("At. ferrooxidans"), lines 240-241(incorrect capitals in species names and lacks italics for the 3 strains), line 243 "feriphilum" should be "ferriphilum", same error in lines 246, 250, 251 (also needs italics), 253, 255; line 316 At. albertensis

lines 321-322 - how was G+C content determined? From sequence information? Please provide a description in the Materials and Methods section.

There are some minor improvements to English usage that should be made. For example, in line 25 of the abstract, "isolate CC was clustered into L. ferriphilum" might read better as "isolate CC was clustered with L. ferriphilum". Another example is line 34 of the Introduction, where there appear to be words missing in the statement "for many years, leptospirilla found to be the dominant...." would read better as "for many years, leptospirilla have been found to be the dominant...". Similar missing words can be found throughout the document, and would benefit from a thorough reading and correction by a native English speaker to improve the readability of the manuscript. This reviewer recognizes none of the authors are primarily English-speakers and in many cases, journals today are permitting some latitude in the quality of the English translation to appear in print.

In line 25, the significant figures for the degree of sequence similarity would be sufficient to estimate as 99.8, rather than "99.80%".

In line 50, page 2, the sentence would read better "The temperature optimum is between 30 to 37C, although many isolated strains are defined as being moderately thermophilic...".

Line 65 would read more simply as "This study" rather than "The presented study".

Line 80 page 2 would read better as "Gram staining was performed by the Huker method...morphology was studied with a Leica DM500...microscope." 

Line 82, section 2.3 heading "Ph" should be changed to "pH"

line 85, rather than "cultivation was done at 150rpm" change to "cultures were agitated at 150 rpm"

line 92 - presumably organic acids were identified by Refractive Index, not "Refractometric Index (RI)" - a general comment about the organic acid analysis: while this is interesting new data, very little discussion is provided about the relevance of these data. Can any comment be made about the major organic acids being tartaric and malic acids? Are these the acids relevant to other studies of EPS in At. ferrooxidans or other chemolithotrophic or heterotrophic species?

In the Materials and Methods section, there is nothing describing the redox potential measurements. What electrode was used, and what was the reference electrode? The question of passivation of pyrite and chalcopyrite with respect to redox potential is critical in chemical or bioleaching of these sulfides. See lines 218-230 where the reference electrode is important to know for the discussion of passivation. For example, was the electrode used Ag/AgCl or SHE? This information is also not available in STable 1.

Line 163 - delete "development" from statement "the phylogenetic (development) tree was built..."

Line 200 - it is unclear in the statement "a highly active resistant microbial consortia" what "resistant" refers to. Heavy metals, solids, high ferric iron or low pH??

Line 279 - change "It is to assume..." to "It is assumed..."

Line 288 - "extra cellular" should be "extracellular"

General comment: the Results section of the manuscript really is a Results and Discussion section. It would be more appropriate to include some of the discussion in the Results section in the Conclusions. As written, the Conclusion section is somewhat disappointing and very brief. This reviewer was expecting to see more discussion of current understanding of the influence of redox potential on chalcopyrite leaching, for example, doi:10.1016/j.mineng.2016.10.003

  • Finally, the authors are asked to review the references for compliance with instructions to authors. Only one DOI is evident.
  •  

Author Response

Reviewer 1

Comments and Suggestions for Authors

The authors are thanked for their interesting contribution describing a new field isolate of Leptospirillum ferriphilum. There are some frequent issues with incorrect capitalization of the species designation ferriphilum through out the manuscript - this should be all lower case, not "Ferriphilum" (note that this error even appears in the title of the manuscript. Please review the text carefully to correct this. There is also inconsistent italicization of the binomial genus and species - for example, in the title and in multiple locations within the abstract (lines 25, 26) where the binomial should be presented in italics. See also line 194, lines 232-233 ("At. ferrooxidans"), lines 240-241(incorrect capitals in species names and lacks italics for the 3 strains), line 243 "feriphilum" should be "ferriphilum", same error in lines 246, 250, 251 (also needs italics), 253, 255; line 316 At. albertensis

Response: Dear Reviewer, first of all, thank you for your helpful comments and constructive suggestions.

All changes in the manuscript are in red.

We corrected the name of “Leptospirillum ferriphilum” in the title, as well as in the whole text. We corrected missing “r” in “ferriphillum” and made italic the name of other strains as well.

lines 321-322 - how was G+C content determined? From sequence information? Please provide a description in the Materials and Methods section.

Response: We have added details on G+C content determination to M&M section.

There are some minor improvements to English usage that should be made. For example, in line 25 of the abstract, "isolate CC was clustered into L. ferriphilum" might read better as "isolate CC was clustered with L. ferriphilum". Another example is line 34 of the Introduction, where there appear to be words missing in the statement "for many years, leptospirilla found to be the dominant...." would read better as "for many years, leptospirilla have been found to be the dominant...". Similar missing words can be found throughout the document, and would benefit from a thorough reading and correction by a native English speaker to improve the readability of the manuscript. This reviewer recognizes none of the authors are primarily English-speakers and in many cases, journals today are permitting some latitude in the quality of the English translation to appear in print.

Response: According to your suggestion we corrected missing words in the whole text.

In line 25, the significant figures for the degree of sequence similarity would be sufficient to estimate as 99.8, rather than "99.80%".

Response: We corrected “99.80%” to “99.8” as you suggested.

In line 50, page 2, the sentence would read better "The temperature optimum is between 30 to 37C, although many isolated strains are defined as being moderately thermophilic...".

Response: We rephrased the sentence according to your suggestion.

Line 65 would read more simply as "This study" rather than "The presented study".

Response: We changed "The presented study" to “This study”.

Line 80 page 2 would read better as "Gram staining was performed by the Huker method...morphology was studied with a Leica DM500...microscope." 

Response: We rephrased the sentence according to your suggestion.

Line 82, section 2.3 heading "Ph" should be changed to "pH"

Response: We corrected “Ph” into “pH”.

line 85, rather than "cultivation was done at 150rpm" change to "cultures were agitated at 150 rpm"

Response: We changed "cultivation was done at 150rpm" to "cultures were agitated at 150 rpm".

line 92 - presumably organic acids were identified by Refractive Index, not "Refractometric Index (RI)" - a general comment about the organic acid analysis: while this is interesting new data, very little discussion is provided about the relevance of these data. Can any comment be made about the major organic acids being tartaric and malic acids? Are these the acids relevant to other studies of EPS in At. ferrooxidans or other chemolithotrophic or heterotrophic species?

Response: Organic acids were determined by HPLC. We added methodology in detail in M&M section.  We also included discussions based on our results and literature data on this issue. We are planing to continue these studies in the near future. These acids are not relevant to EPS of chemolithotrophic bacteria.

In the Materials and Methods section, there is nothing describing the redox potential measurements. What electrode was used, and what was the reference electrode? The question of passivation of pyrite and chalcopyrite with respect to redox potential is critical in chemical or bioleaching of these sulfides. See lines 218-230 where the reference electrode is important to know for the discussion of passivation. For example, was the electrode used Ag/AgCl or SHE? This information is also not available in STable 1.

Response: We have added Ag/AgCl in the STable 1. We have also added information about redox potential measurements in M&M section.

Line 163 - delete "development" from statement "the phylogenetic (development) tree was built..."

Response: We deleted "development" as suggested. 

Line 200 - it is unclear in the statement "a highly active resistant microbial consortia" what "resistant" refers to. Heavy metals, solids, high ferric iron or low pH??

Response: This statement was based on our previous paper [Ref. 32]. Nevertheless, we have deleted it.

Line 279 - change "It is to assume..." to "It is assumed..."

Response: We changed "It is to assume..." to "It is assumed...".

Line 288 - "extra cellular" should be "extracellular"
Response: we corrected the misspelling.

General comment: the Results section of the manuscript really is a Results and Discussion section. It would be more appropriate to include some of the discussion in the Results section in the Conclusions. As written, the Conclusion section is somewhat disappointing and very brief. This reviewer was expecting to see more discussion of current understanding of the influence of redox potential on chalcopyrite leaching, for example, doi:10.1016/j.mineng.2016.10.003

Response: We rewrote conclusion. In this work we aimed to carry out comparative study on bioleaching of pyrite and chalcopyrite by pure and mixed cultures.

We will take into consideration your helpful suggestions to study influence of redox potential on chalcopyrite leaching in our further work. 

Finally, the authors are asked to review the references for compliance with instructions to authors. Only one DOI is evident.

Response: We checked the references attentively and added some new literature. As DOI seems not obligatory, we have deleted it.

Reviewer 2 Report

This manuscript is the continuation of the previous work on the strain Leptospirillum ferriphilum CC [reference 43 in the text of the manuscript]. However, some parts need explanations about the novelty or exclusion from the text since they seem to be the repetition of the previous work. Overall, the manuscript needs a substantial revision before it can be processed further.

Why is this strain new? Please delete “new” everywhere in the text. A paper about this strain has already been published by you [reference 43]. Please exclude all data previously published from the results section, such as Section 3.4, the results of which were not obtained in the present study. Please see the specific comments below.

Please provide a clear discussion of your results together with the previously published data by you and other authors and enrich the text with relevant references. There is no Discussion section in the manuscript. If instead of the Results section a Results and Discussion section was meant, anyway, a clear discussion of the results is needed. The article is devoted to the strain, which is very close to the known ones. Therefore, please compare your strain to the phylogenetically close strains of this species and specify what makes the strain CC unique from them. Enrich your manuscript with the data on the leaching of other sulfide minerals (chalcopyrite, pyrite, sphalerite, etc.) by L. ferriphilum (not only L. ferrooxidans), especially in mixed cultures. Compare with the strains that are phylogenetically close to your isolate if available in the literature.

See also the specific comments below.

Section 3.6.3.” Biooxidation of Tandzut Gold-Bearing Pyritic Ore” is not a logical continuation of Results. Please explain why you chose this particular ore. What was the mineral and chemical composition of this ore? What were the characteristics of this sample?

Conclusions need corrections according to the modified text of other sections.

Specific Comments:

1.    Title:

The strain is not new. Please change the title to the one that refers to the content of the present paper. The title now states only the existence of the strain.

2.    Line 22 (Abstract): “Strain CC was an obligate chemolithoautotroph”. Did you check its growth on organic media in this work that this conclusion is in the abstract? Or is it just based on the known facts about bacteria of the genus Leptospirillum? Please delete this phrase.

3.    Lines 26-28 (Abstract): The data on the bioleaching of pyrite and chalcopyrite, together with the extraction of iron and copper from these minerals, were provided in your previous work [43]. If it is not so, please explain what was novel in these experiments.

4.    Lines 28-29 (Abstract): “:..contribute to the iron cycling as well as to the acid mine drainage patterns in the local area”. It is not discussed anywhere in the text. Please add this part to the Results and Discussion section if you mention it in the abstract.

5.    In Abstract, no information about results on the bioleaching with mixed cultures is mentioned. It should be enriched with these data.

6.    L. 39: “Besides L. ferrooxidans as well as L.ferriphilum…” is an unclear sentence. Please correct.

7.    L. 72-78: Why do you mention the isolation of this strain from the enrichment culture in Materials and Methods if this strain was isolated in the previous work [43]? Was it isolated again from the material? Please correct the text of Section 2.1. In addition, enrich this section with information on the cultivation of other bacteria used in experiments.

8.    L. 85-86: These growth ranges in Materials and Methods are different from those provided in the results. They should be the same in both sections.

9.    L. 88: Why are colloidal polysaccharides mentioned here?

10.  What was the mineralogical composition of the chalcopyrite sample? It is seen from the chemical composition and results of the leaching that it is not a pure chalcopyrite.

11.  What was the mineral and chemical composition the Tandzut gold-bearing pyritic ore? What were the characteristics of this sample? No information is provided in Materials and Methods.

12.  Table 1: No units are given, is it percentage (wt%)?

13.  Line 111: In the text of Results, the temperature is 30 °C when mixtures of bacteria are used.

14.  Information about the controls that you show in the Results should be added to the Materials and Methods section.

15.  Lines 125-134: Is it a novel procedure used for L. ferriphilum? Why the previously isolated strain was isolated again? Please see 7.

16.  Line 137: Should there be Figure 1 instead of Fig. 3?

17.  Line 147: Why Figure S2 is mentioned here?

18.  Line 150: Figure 2: Please specify what you mean by iron oxidation activity. Maximum concentration of ferric iron during the growth of the strain? At pH values higher than 2, Fe3+ begins to precipitate, and especially at pH 3. Therefore, the data for Figure 2b do not correctly show the effect of pH on Fe2+ oxidation. Please specify in the legend of Figure 2 at what pH was the growth measured in (a) and at what temperature was the optimum pH tested in (b).

19.  Line 152: This part (Section 3.4) should be excluded. It does not provide the results of the present study.

20.  Figure 3 (line 169): The title needs correction. It is not the phylogenetic tree of only the strain CC.

21. Results section. Please compare your strain with the strains that the isolate CC is clustered together with, according to the data on these strains in the literature. Add the references that are available. What makes the isolate CC unique? In Table 2, I recommend to add the sources of the isolation of the strain CC and other strains with references and probably some other characteristics to compare them. Only one strain is the type strain (P3a), therefore, the title of Table 2 is wrong. Correct please.

22. Section 3.6: The data on the ability of this strain to oxidize pyrite and chalcopyrite have been reported in the previous paper [reference 43: Table 1 and Figure 1 (red lines)]. The strain is the same in the current work but the data vary. Please delete this part or specify what is novel in this part and explain the varying figures in two papers with the same strain.

23. Lines 185-193: In reference 43, not only the association but also pure cultures (including the pure culture of the isolate CC) were used and, therefore, these data have been previously reported. Please see the previous comments and explain what was new in this study regarding the bioleaching with the pure culture of the isolate CC compared to the previous paper.

24. Lines 204-207: The sentence is unclear. Please correct.

25. Figures 7, 8, 9: If the temperature was 30 °C, correct this also in Materials and Methods.

26. Line 219: Correct “amounts (numbers?).

27. Section 3.6.2 and Table 3: Did you measure only the copper concentration in the liquid phase? Do you have data on the content of copper in leach residues? Statistical errors of values are not given in tables. Why the difference between extracted Fe and Cu from CuFeS2 is almost 2 times?

28. Can you explain the presence of copper in controls at 40 and 30 degrees and at the beginning of leaching experiments in the medium?

29. Is Equation 3 equalized?

30. Tables 4 and 5; line 289. How the data of Tables 4 and 5 on organic acids are connected with EPS? Please explain the aim to measure these acids that are related to metabolism pathways and discuss the data obtained by you and the literature data to compare the results obtained.

31. Section 3.6.3: This part does not seem logical in this manuscript. Please provide more data and explanations why it was included here.

32. L. 315: What is meant by “the degree of leaching of the Tandzut ore?” Please specify.

33.  Table S1. From what material are data on the iron leaching shown? The title of Table S1 should be specified.

34. Figure S2: Conditions of growth should be specified.

35. Please correct all errors and typos throughout the text, including the spelling of the microbial names.

Author Response

Reviewer 2

Comments and Suggestions for Authors

This manuscript is the continuation of the previous work on the strain Leptospirillum ferriphilum CC [reference 43 in the text of the manuscript]. However, some parts need explanations about the novelty or exclusion from the text since they seem to be the repetition of the previous work. Overall, the manuscript needs a substantial revision before it can be processed further.

Why is this strain new? Please delete “new” everywhere in the text. A paper about this strain has already been published by you [reference 43]. Please exclude all data previously published from the results section, such as Section 3.4, the results of which were not obtained in the present study. Please see the specific comments below.

Response: Dear Reviewer, first of all, thank you for your helpful comments and constructive suggestions.

All changes in the manuscript are in red.

We have deleted the section.

Please provide a clear discussion of your results together with the previously published data by you and other authors and enrich the text with relevant references. There is no Discussion section in the manuscript. If instead of the Results section a Results and Discussion section was meant, anyway, a clear discussion of the results is needed. The article is devoted to the strain, which is very close to the known ones. Therefore, please compare your strain to the phylogenetically close strains of this species and specify what makes the strain CC unique from them. Enrich your manuscript with the data on the leaching of other sulfide minerals (chalcopyrite, pyrite, sphalerite, etc.) by L. ferriphilum (not only L. ferrooxidans), especially in mixed cultures. Compare with the strains that are phylogenetically close to your isolate if available in the literature.

Response: We have added references in Table 2.

See also the specific comments below.

Section 3.6.3.” Biooxidation of Tandzut Gold-Bearing Pyritic Ore” is not a logical continuation of Results. Please explain why you chose this particular ore. What was the mineral and chemical composition of this ore? What were the characteristics of this sample?

Response: We deleted the descriptions regarding biooxidation of Tandzut gold-bearing pyritic ore.

Conclusions need corrections according to the modified text of other sections.

Response: We have modified Conclusion section accordingly.

Specific Comments:

  1. Title:

The strain is not new. Please change the title to the one that refers to the content of the present paper. The title now states only the existence of the strain.

Response: We changed the title. It reads now ‘Bioleaching of Sulfide Minerals by Leptospirillum ferriphilum CC from Polymetallic Mine (Armenia)’

  1. Line 22 (Abstract): “Strain CC was an obligate chemolithoautotroph”. Did you check its growth on organic media in this work that this conclusion is in the abstract? Or is it just based on the known facts about bacteria of the genus Leptospirillum? Please delete this phrase.

Response: We didn’t observe the growth of L.ferriphilum CC in MAC medium supplemented with glucose, saccharose, some organic and amino acids, as well as yeast extract both in mixotrophic and heterotrophic conditions.

  1. Lines 26-28 (Abstract): The data on the bioleaching of pyrite and chalcopyrite, together with the extraction of iron and copper from these minerals, were provided in your previous work [43]. If it is not so, please explain what was novel in these experiments.

Response: In our previous work [Ref. 32] comparative activities of L. ferriphilum CC and At. ferrooxidans Ksh, as well as their association in bioleaching of pyrite and chalcopyrite were studied. While in this work we used L.feriiphilum CC in association with sulfur oxidizing bacterium At. Albertensis SO-2 and heterotrophic bacterium Acidocella sp. RBA. So, comparative studies of pyrite and chalcopyrite bioleaching by pure culture of L. ferriphillum CC and its association with sulfur oxidizing bacteria and heterotrophic bacteria, as well as At. ferrooxidans ZnC were carried out.  

  1. Lines 28-29 (Abstract): “:..contribute to the iron cycling as well as to the acid mine drainage patterns in the local area”. It is not discussed anywhere in the text. Please add this part to the Results and Discussion section if you mention it in the abstract.

Response: We deleted the sentence from Abstract and added the mentioned sentence in Conclusion section as suggested.

  1. In Abstract, no information about results on the bioleaching with mixed cultures is mentioned. It should be enriched with these data.

Response: We have added results on bioleaching of pyrite and chalcopyrite with mixed cultures in abstract.

  1. L. 39: “Besides L. ferrooxidans as well as L.ferriphilum…” is an unclear sentence. Please correct.

Response: We rephrased the sentence as suggested. (Line 42)

  1. L. 72-78: Why do you mention the isolation of this strain from the enrichment culture in Materials and Methods if this strain wasisolated in the previous work [43]? Was it isolated again from the material? Please correct the text of Section 2.1. In addition, enrich this section with information on the cultivation of other bacteria used in experiments.

Response: We rewrote Section 2.1.

  1. L. 85-86: These growth ranges in Materials and Methods are different from those provided in the results. They should be the same in both sections.

Response: We corrected temperature ranges for growth of L. ferriphilum CC to 25-50°C. (Line 96)

  1. L. 88: Why are colloidal polysaccharides mentioned here?

Response: We corrected the section.

  1. What was the mineralogical composition of the chalcopyrite sample? It is seen from the chemical composition and results of the leaching that it is not a pure chalcopyrite.

Response: Chalcopyrite samples were provided by the Shamlugh Mining company and it’s purity was higher than 98%?.

  1. What was the mineral and chemical composition the Tandzut gold-bearing pyritic ore? What were the characteristics of this sample? No information is provided in Materials and Methods.

Response: We have deleted the descriptions on Tandzut gold-bearing pyritic ore.

  1. Table 1: No units are given, is it percentage (wt%)?

Response: We have added unites in the caption of Table 1 (wt%).

  1. Line 111: In the text of Results, the temperature is 30 °C when mixtures of bacteria are used.

Response: As At. ferrooxidans ZnC and RBA are mesophilic bacteria, comparative studies on bioleaching of minerals with pure and mixed cultures were carried out at 35oC that is closer to optimal growth temperatures of all bacteria used. We did revisions in Material and Methods part.

  1. Information about the controls that you show in the Results should be added to the Materials and Methods section.

Response: We have added information about the controls to the Materials and Methods section. (Lines 137-139)

  1. Lines 125-134: Is it a novel procedure used for L. ferriphilum? Why the previously isolated strain was isolated again? Please see 7.

Response: We have removed relevant descriptions as suggested.

  1. Line 137: Should there be Figure 1 instead of Fig. 3?

Response: Thank you, we corrected the mistake.

  1. Line 147: Why Figure S2 is mentioned here?

Response: We corrected the mistake; it shout be replaced by ‘Fig. 1’.

  1. Line 150: Figure 2: Please specify what you mean by iron oxidation activity. Maximum concentration of ferric iron during the growth of the strain? At pH values higher than 2, Fe3+ begins to precipitate, and especially at pH 3. Therefore, the data for Figure 2b do not correctly show the effect of pH on Fe2+ oxidation. Please specify in the legend of Figure 2 at what pH was the growth measured in (a) and at what temperature was the optimum pH tested in (b).

Response: According to your comments we added the experimental conditions in caption of Fig. 2 (pH for Fig 2a) and (temperature for Fig. 2b).

  1. Line 152: This part (Section 3.4) should be excluded. It does not provide the results of the present study.

Response: We kept only the conclusion from previous work concerning kinetic parameters of bacteria. We deleted subtitle 3.4 and joined to the Section 3.2.

  1. Figure 3 (line 169): The title needs correction. It is not the phylogenetic tree of only the strain CC.

Response: We rewrote the title of figure legend of Figure 3 as follows: Phylogenetic position of strain CC.

  1. Results section. Please compare your strain with the strains that the isolate CC is clustered together with, according to the data on these strains in the literature. Add the references that are available. What makes the isolate CC unique? In Table 2, I recommend to add the sources of the isolation of the strain CC and other strains with references and probably some other characteristics to compare them. Only one strain is the type strain (P3a), therefore, the title of Table 2 is wrong. Correct please.

Response: We rewrote the title in Table 2. Identity of 16s rRNA of isolated Leptospirillum sp. CC with other L.ferriphilum strains. We have added relevant references in Table 2.

  1. Section 3.6: The data on the ability of this strain to oxidize pyrite and chalcopyrite have been reported in the previous paper [reference 43: Table 1 and Figure 1 (red lines)]. The strain is the samein the current work but the data vary. Please delete this part or specify what is novel in this part and explain the varying figures in two papers with the same strain.

Response: Obtained data on bioleaching of minerals by L. ferriphilum in this paper and the previous work vary because the experimental conditions were different.

In this study -pulp density (PD) and initial pH were 4% and pH 1.5, respectively, t-35°C, inoculum- 5% about 107 - 108 cells/ml. In the previous work PD was 10%, pH-1.7, t-37°C, inoculum- 2x108 cells/ml.

  1. Lines 185-193: In reference 43, not only the association but also pure cultures (including the pure culture of the isolate CC) were used and, therefore, these data have been previously reported. Please see the previous comments and explain what was new in this study regarding the bioleaching with the pure culture of the isolate CC compared to the previous paper.

Response: In our previous work [ref. 32] comparative activities of L. ferriphilum CC and At. ferrooxidans Ksh, as well as their association in bioleaching of pyrite and chalcopyrite were studied. While in this work we used L.feriiphilum CC in association with sulfur oxidizing bacterium At. Albertensis SO-2 and heterotrophic bacterium Acidocella sp. RBA. So, comparative studies of pyrite and chalcopyrite bioleaching by pure culture of L. ferriphillum CC and its association with sulfur oxidizing bacteria and heterotrophic bacteria, as well as At. ferrooxidans ZnC were carried out.  

  1. Lines 204-207: The sentence is unclear. Please correct.

Response: We deleted the sentence.

  1. Figures 7, 8, 9: If the temperature was 30 °C, correct this also in Materials and Methods.

Response: We corrected the numbering of figures. Temperature in figures 6, 7 and 8 should be 35°C. We did appropriate correction in Material and Methods part. As At. ferrooxidans ZnC and RBA are mesophilic bacteria, comparative studies on bioleaching of minerals with pure and mixed cultures were carried out at 35°C that is closer to optimal growth temperatures of all bacteria used.

  1. Line 219: Correct “amounts (numbers?)”.

Response: We corrected it as suggested.

  1. Section 3.6.2 and Table 3: Did you measure only the copper concentration in the liquid phase? Do you have data on the content of copper in leach residues? Statistical errors of values are not given in tables. Why the difference between extracted Fe and Cu from CuFeSis almost 2 times?

Response: Yes, we measured only copper, Fe2+, Fe3+ and total iron in liquid phase. Unfortunately, we have no data on content of copper in leaching residues.

  1. Can you explain the presence of copper in controls at 40 and 30 degrees and at the beginning of leaching experiments in the medium?

Response: Chalcopyrite is acid soluble. That’s why some abiotic dissolution of chalcopyrite could occur in acidic solution as a result of protonic attack.

  1. Is Equation 3 equalized?

Response: We equalized Equation 3.

  1. Tables 4 and 5; line 289. How the data of Tables 4 and 5 on organic acids are connected with EPS? Please explain the aim to measure these acids that are related to metabolism pathways and discuss the data obtained by you and the literature data to compare the results obtained.

Response: Our previous work showed that the cell lysis products (CLs) of L. ferriphilum and At. ferrooxidans ZnC can inhibit oxidation of Fe2+ by these two strains (Vardanyan et al., 2017). Taking into consideration the above mentioned we carried out analysis of culture liquid resulted from grown of L. ferriphilum and At. ferrooxidans ZnC in the medium with Fe2+ using HPLC. As a result, we detected some organic acids in the culture liquid of these bacteria. On the other side some studies have shown that the co-culture of heterotrophs with chemolithotrophic microorganisms leads to higher microbial activity compared to the presence of only chemolithotrophs. Berthelot et al. (1997), Johnson and Roberto (1997) and Liu et al. (2011) have proposed that the presence of heterotrophs, such as A. acidophilum, can improve metal bioleaching by biodegrading organic matter and thus detoxify the growth environment for other chemolithotrophic acidophiles. Vardanyan et al. (2017) showed that Acidocella sp. RBA can detoxify the environment from CLs products and promote chemolithotrophic growth.

  1. Section 3.6.3: This part does not seem logical in this manuscript. Please provide more data and explanations why it was included here.

Response: We have deleted Section 3.6.3 according to your suggestion.

  1. L. 315: What is meant by “the degree of leaching of the Tandzut ore?” Please specify.

Response: We have deleted the section.

  1. Table S1. From what material are data on the iron leaching shown? The title of Table S1 should be specified.

Response: We corrected the title of Table S1.  

  1. Figure S2: Conditions of growth should be specified.

Response: We have added information regarding growth condition of SEM images. 

  1. Please correct all errorsand typos throughout the text, including the spelling of the microbial names.

Response: We have corrected spelling of the microbial names.

Round 2

Reviewer 2 Report

The authors have considerably improved their manuscript. Most comments have been addressed.

However, two important issues still need clarification (comment no. 27 in my previous review). If according to the authors’ reply, the chalcopyrite sample is pure, the content of Cu and S should be equal. Instead, the content of sulfur is approx. 8% higher (Table 1). In Table 3, the difference between extracted Fe and Cu (from CuFeS2) is almost two times. This inconsistency should be explained. Otherwise, these data cannot be considered correct.

Other comments:

Table 2 (l. 180): Please correct Ref. [50], which has been added to the revised manuscript, in Table 2 and References. The order of authors is incorrect (the first author is different), and the journal name is missing (Microbiology)  https://doi.org/10.1134/S0026261718030086.

l. 94: In Methods, the pH range is 0.5–2.5. In the Results (l. 160), the pH range is 0.5–3.0 (Figure 2, Effect of temperature (a) (at pH 1.6) and pH (b) (at temperature 40°C) on iron oxidation by L. ferriphillum CC). Please remove pH 3.0 from Figure 2. At pH 3.0, iron precipitates, influencing the data on iron oxidation based on the concentration of Fe3+ (comment no. 18 in my first review). L. ferriphillum is misspelled (l. 161).

Author Response

Reviewer 2

The authors have considerably improved their manuscript. Most comments have been addressed.

However, two important issues still need clarification (comment no. 27 in my previous review). If according to the authors’ reply, the chalcopyrite sample is pure, the content of Cu and S should be equal. Instead, the content of sulfur is approx. 8% higher (Table 1). In Table 3, the difference between extracted Fe and Cu (from CuFeS2) is almost two times. This inconsistency should be explained. Otherwise, these data cannot be considered correct.

Response: Dear Reviewer thank you for your additional comments.

New changes in the manuscript are in blue.

We misspelled the content of S in chalcopyrite in Table 1. The content S in chalcopyrite should be 33.8.

As for Table 3: Taking into consideration that the amounts of Cu and S in CuFeS2 should be equivalent, it can be supposed that chalcopyrite contained a small amount of other copper mineral (for example enargite (Cu3AsS4)) resulted in higher content of sulfur. The mentioned reason can explain the difference observed between the amounts of extracted Fe and Cu in Table 3.

Anyway, we added our assumption below Table 1.

Other comments:

Table 2 (l. 180): Please correct Ref. [50], which has been added to the revised manuscript, in Table 2 and References. The order of authors is incorrect (the first author is different), and the journal name is missing (Microbiology) https://doi.org/10.1134/S0026261718030086.

Response: We revised Ref. 50 in Table 2 and in the Reference list.

I.94: In Methods, the pH range is 0.5–2.5. In the Results (l. 160), the pH range is 0.5–3.0 (Figure 2, ‘Effect of temperature (a) (at pH 1.6) and pH (b) (at temperature 40°C) on iron oxidation by L. ferriphillumCC’). Please remove pH 3.0 from Figure 2. At pH 3.0, ironprecipitates, influencing the data on iron oxidation based on the concentration of Fe3+ (comment no. 18 in my first review). L. ferriphillum is misspelled (l. 161).

Response: We removed pH 3.0 from Fig. 2. We corrected also misspelling of “ferriphilum” in the whole text.
